# Review of Research on Urban Social Space and Sustainable Development

**Jian Feng ***  and **Huali Hou**

College of Urban and Environmental Sciences, Peking University, Beijing 100871, China; 2201213393@pku.edu.cn
* Correspondence: fengjian@pku.edu.cn

**Abstract:** Urban social space and sustainable urban development are both prominent areas of research in urban studies. The development of a city is closely tied to the development of its social space. The level of sustainable development in a city can be assessed by examining the evolution of its urban social space. Therefore, the two are highly interconnected in a close relationship. However, the social dimension of sustainable development has always received the least attention compared to the economy and the environment. Therefore, this paper examines urban transformation and urban spatial structure, social composition, the activity space and living space of urban residents, the social integration of urban residents in urban communities, and urban planning. This paper provides a comprehensive literature review of research on urban social space and sustainable development. On the one hand, this literature review thoroughly examines the correlation between urban social space and sustainable urban development. On the other hand, it broadens the perspectives of urban research, highlights the key role of social dimensions in sustainable urban development, and helps to draw the attention of academics to this topic. In addition, this literature review may provide policymakers with more comprehensive information on urban social space and sustainable development.

**Keywords:** urban social space; sustainable development; literature review

## 1. Introduction

In 1987, members of the World Commission on Environment and Development, headed by Mrs. Brundtland, formally proposed the concept of "sustainable development" in their report "Our Common Future" [1]. Since then, the concept has been widely applied to environmental, economic, and social research fields and has now become a ubiquitous term. Simply put, sustainability is a concept that addresses questions and concerns regarding what defines a quality life in a specific location and era [2]. This involves evaluating the utilization and preservation of available resources to support the future needs of both human populations and the environment. The implementation of sustainable practices within a system is crucial to minimize the negative impacts on both society and the natural world. Today, an increasing number of people live and work in urban areas, and the majority of economic, political, and social activities take place in cities [3]. In 2008, the global urban population exceeded the rural population for the first time in history [4]. It is estimated that by 2050, the proportion of the global urban population will reach 68% [5]. In the context of global urbanization, the United Nations has incorporated sustainable urban development into the 2030 Agenda, specifically in Article 11 of the Sustainable Development Goals (SDGs): "Make cities and human settlements inclusive, safe, resilient, and sustainable" [4]. The concept of "sustainable development" has gradually become the cornerstone of many urban development plans and strategies worldwide [6].

Society, environment, and economy are the three pillars of urban sustainability research discourse [7]. The theory of sustainable development fully emphasizes the coordination and sustainability among the three pillars [8].

Consequently, the academic community has shown significant interest in the sustainable development of the environment [9]. Rapid urban development has caused extensive ecological damage, harming the resources on which human beings depend for survival, such as water, the atmosphere, and soil [8]. Therefore, the protection and management of the ecological environment are the focus and hotspot of the study of environmental sustainability. Programs such as developing a low-carbon and green economy [10], promoting recycling and urban waste classification [11], and urban planning based on an ecological perspective [12] have been proposed. Research on measuring the effectiveness of environmental protection and governance policies, such as evaluating ecological sustainability policies [9] and assessing the sustainability of urban ecosystems [13], is also a crucial area of study in environmental sustainability research.

Economic sustainability focuses on the maintenance of capital resources, including manufacturing capital, natural capital, and financial capital [14]. It emphasizes that economic growth should not jeopardize the economic prospects of future generations [15]. Existing studies focus on the factors that impact economic sustainability, such as the level of innovation [16] and the digital economy [17]. In addition, studies on economic sustainability are often intertwined with ecology, with a focus on topics such as the ecological impact on sustainable economic development [18] and the environmental assessment of economic development [19].

Urban social sustainability has been the least defined [20] and least focused [21] dimension. Social sustainability focuses on social well-being, emphasizing social equity, social cohesion, etc., and focusing on a variety of topics such as the basic needs of people's daily lives (e.g., housing, education, transportation, etc.), cultural integration, and other topics, with an emphasis on the goals of addressing social inequality and improving the well-being of residents [7,20,21].

The development of cities depends on the development of their social space [6], and groups in different regions and social contexts perceive, experience, and use cities in very different ways, with very different visions of sustainable development. That is, sustainable development itself is spatially related and occurs in a specific social context [22] as a complex and socio-spatially sensitive phenomenon [2]. The socio-spatial dimension is therefore extremely important in sustainability research and it is necessary to develop an understanding about the two.

Since the French sociologist E. Durkheim first proposed the concept of social space in the late 19th century, scholars have paid attention to this topic from the perspectives of geography, sociology, architecture, philosophy, and other disciplines [23]. Urban social space is a concentrated representation of various social activities and social relations in the city [24], and it is important to study social space to understand urban spatial structure and guide urban planning practices in a deeper way [25]. The development status and trend of social space are important aspects for measuring the sustainable development status of cities. With the deepening of urbanization, modernization, and globalization, global cities are constantly presenting increasingly subdivided, fragmented, and polycentric social spaces [26]. In this context, the sustainability of urban social space as a fundamental component of sustainable development has received increasing attention [7]. It has also emerged as an important area of research and practice in urban planning policies in both developed and developing countries [20].

Overall, the study of urban socio-spatial and sustainable development focuses on both socio-spatial elements (e.g., urban socio-spatial structure, living space, and activity space) and social sustainability (e.g., community integration, improvement of residents' well-being, reduction in social inequality, etc.). Therefore, a review of the research on urban social space and sustainable development can draw scholars' attention to the intersection of these two fields and provide a reference for urban sustainable development planning (Figure 1).

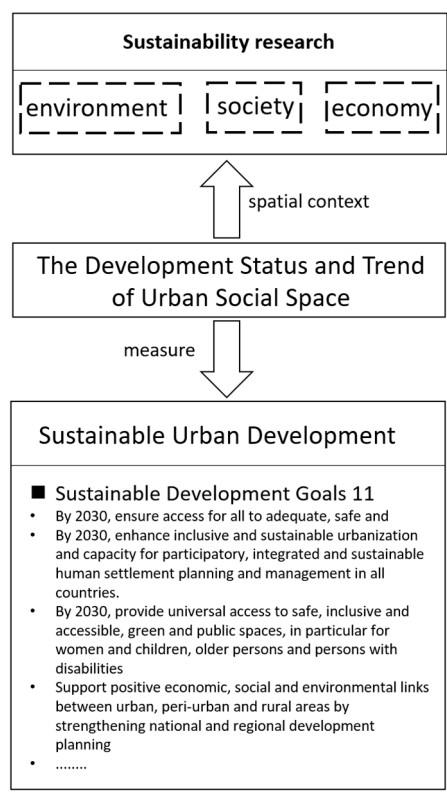

**Figure 1.** The strong link between urban social space and sustainable development [4,6–8,22,26].

To provide readers with a clear roadmap, this review paper is carefully structured as follows: Section 1 not only introduces the complex and essential connection between urban social space and sustainable development but also highlights the importance of exploring this interdependent relationship. Sections 2–5 meticulously navigate the trajectory of research progress, covering various topics. These chapters explore everything from the macroscopic to the microscopic, from theoretical frameworks to empirical substantiation, and academic inquiry to practical implementation. These chapters delve into four distinct facets of urban social space and sustainable development, providing a comprehensive perspective. Firstly, it explores the transformation and restructuring of urban social space. Secondly, it examines the social composition of urban community residents and social integration. Thirdly, it analyzes the activity spaces and living spaces of urban residents. Lastly, it offers an in-depth review of urban planning in the context of sustainable development. The last Section consists of the conclusion and discussion of this paper.

## 2. Transformation and Restructuring of Urban Social Space

With the deepening of urbanization, modernization, and globalization, spatial reconfiguration has become one of the most significant changes in the transformation of urban societies during the transition period, and the transformation of urban societies is manifested in spatial transformation [27]. On the one hand, urban social space provides a place for elements such as power, interests, and ideas to take place [28]. On the other hand, it also obscures and solidifies deep-seated social problems, such as social stratification, power conflicts, and competing interests, which are hidden behind phenomena like urban expansion, population movement, and industrial restructuring [29]. These social problems seriously hinder the sustainable development of cities. Therefore, it is necessary to study the transformation and reconstruction of urban social space, identify its characteristics, discover the existing problems, and propose effective solutions. This has both theoretical and practical significance in achieving sustainable urban development.

*2.1. Theories of Urban Socio-Spatial Transformation and Reconfiguration*

A review of existing studies reveals that there are different theoretical perspectives on the transformation and reconstruction of urban social space (Table 1). Early on, the Chicago School studied the socio-spatial structure of the city. The urban ecology theory that they established regarded the whole city as an organism, in which individuals in the city achieve symbiosis through division of labor and cooperation, groups compete for limited land resources, and the socio-spatial structure of the city depends on the results of various ecological processes [30]. In the theory of urban ecology, the socio-spatial characteristics of people are interpreted as a reflection of inter-group struggle and cooperation [31]. To reflect the urban socio-spatial characteristics of Chicago, Burgess constructed the concentric circle model, which divides the various functional areas in the city through a series of concentric circles [32]. Later, scholars improved on the concentric circle model and proposed the sector model [33] and the multiple nuclei model [34]. Compared with the concentric circle model, the sector model reflects the outward extension of the city, while the multiple nuclei model emphasizes the polycentricity of the city [35]. The theory has been widely influential since its introduction, and many subsequent studies of urban socio-spatial structure by scholars have been based on critiques or reflections of the theory.

**Table 1.** Main theoretical perspectives in the study of the transformation and restructuring of the urban socio-spatial structure.

| Theoretical Perspectives | Description |
| --- | --- |
| Chicago School and Urban Ecology | - The Chicago School's urban ecology theory views the city as an organism with socio-spatial structures shaped by ecological processes. |
| | - Burgess's concentric circle model divides the city into functional areas through concentric circles. |
| | - Later models, like the sector and multiple nuclei models, expanded upon and refined the original theory. |
| Dialectical Analysis | - Analyzes urban socio-spatial transformation from a metaphysical and historical perspective. |
| Feminist–Materialist Theory | - Focuses on anti-racist feminist–materialist urban theory, emphasizing care, the common, and the collective in urban socio-spatial transformation. |
| Neoliberalism | - Analyzes urban socio-spatial transformation within the context of neoliberalism. |
| Social Engineering | - Examines government-dominated cities where social engineering ideas and an "engineering type of mind" play a significant role in transformation. |

The analysis of urban socio-spatial transformation based on the philosophical metaphysical level of "social-spatial" dialectic is an important entry point, and we can also see the process of reconstructing urban socio-spatial from the metaphysical history of urban development [29]. Feminist scholars, on the other hand, propose to adopt an anti-racist feminist–materialist urban theory to analyze the socio-spatial transformation and reconstruction of the city, focusing on the caring, the common, and the collective [36]. With the widespread practice of neoliberalism around the world, the study of urban socio-spatial transformation and reconstruction from the context of neoliberalism has been advocated [37,38]. For countries with high government dominance, institutions play a relatively large role in urban socio-spatial transformation and reconstruction; social engineering ideas are widely used in social construction, and an "engineering type of mind" is an important perspective to study the socio-spatial transformation and reconstruction of government-dominated cities [27]. In addition, the rapid development of transportation has reduced barriers to population mobility, resulting in various forms of socio-spatial reconfiguration such as gentrification [39], class transformation [40], and spatial segregation of migrants [41]. Therefore, it is also relevant to study the urban socio-spatial transformation and reconfiguration from the perspective of population migration.

### 2.2. Empirical Studies of Urban Socio-Spatial Transformation and Reconfiguration

The empirical study of urban socio-spatial transformation and reconfiguration focuses on four aspects. The first objective is to discuss the characteristics of urban socio-spatial reconfiguration in the context of specific cases from different scales such as international [37,42], urban [43–45], and community/specific intra-urban spaces [46–49]. The second is to analyze the mechanisms that influence urban socio-spatial reconfiguration [42,50,51]. The third is to analyze the effects of urban socio-spatial transformation and reconfiguration [52–54]. Finally, the discussion of how to address the negative impacts arising from urban socio-spatial transformation and reconfiguration to achieve sustainable urban development [55–57] is also a topic of interest (Table 2).

**Table 2.** Summary of Empirical Studies on Socio-Spatial Transformation and Restructuring.

| Empirical Studies | Description |
| --- | --- |
| **Characteristics of Transformation and Restructuring** | - Examines urban socio-spatial reconfiguration at various scales (international, urban, and intra-urban spaces) in specific cases. |
| **Mechanisms of Transformation and Restructuring** | - Analyzes the mechanisms influencing urban socio-spatial reconfiguration, including administrative, market, and social forces. |
| **Effects of Transformation and Restructuring** | - Investigates the effects of urban socio-spatial transformation, including its social and environmental impacts. |
| **Addressing Negative Impacts** | - Discusses strategies to mitigate the negative impacts of urban socio-spatial transformation for sustainable urban development. |

In exploring the characteristics of urban socio-spatial transformations and reconfigurations, scholars have enriched this issue from different scales. On the international scale, Ghahremani et al. selected four global cities (Istanbul, Turkey; Shanghai, China; Singapore; and Tokyo, Japan) for comparative analysis [42]. Apostolopoulou analyzed the trajectory of urban social space transformation and reconstruction driven by the Belt and Road Initiatives [37]. On the urban scale, there is a considerable amount of relevant research. In St. Petersburg, socio-spatial reorganization and urban morphological changes are always present as two dimensions of the post-socialist urban transformation [43]. In Asian cities, the contradictions in the process of urban socio-spatial transformation and reconfiguration are particularly pronounced [58]. Taking China as an example, China has experienced an important stage of transition from a planned economy to a socialist market economy [26]. The reconfiguration of intra-city space in China during the transition period follows an important pattern of transformation from the homogeneous spatial structure of the planned economy to the heterogeneous spatial structure of the market economy [44]. Among them, the socio-spatial transformation and reconfiguration characteristics of special cities such as Beijing, the capital of China [23,45], and resource-based cities [25] have received much attention.

In discussing the factors influencing urban socio-spatial transformation and reconfiguration, scholars have given different explanations in the context of specific cases. In China, the impetus and interaction of administrative, market, and social forces are the driving forces that make the urban socio-spatial structure continue to evolve [45]. In Chinese tourist cities, natural resource endowment, government policy guidance, market orientation, and participation of social agents are the main factors influencing their socio-spatial transformation and reconfiguration [53]. Placing cities in the context of globalization and urban competition, potential factors influencing the socio-spatial transformation and reconfiguration of cities include environmental and social sustainability, the application of modern technologies, service-based economy, and knowledge-based industries [42]. In addition,

the impact of specific factors, such as the digital economy [50] and specific institutions [37], on urban socio-spatial transformation and reconfiguration has also received attention.

The rapid and large-scale process of urban socio-spatial transformation and reconfiguration has produced multiple social impacts. Among them, informal space is one of the most prominent issues. Rapid urbanization and urban transformation processes have resulted in a large proportion of the world's population living in informal spaces in cities [59]. Informal spaces are of great concern globally [60], and this phenomenon is particularly prevalent in cities in the global south [61]. Shantytowns, for example, have received much attention as typical informal spaces [61–63]. Informal space has the attributes of mixed-use development, high density, and compactness, compared to formal space, which provides residents with functional and personalized polymorphic spaces, but the space is subject to weak regulation and social problems such as fire risk, poverty, and anti-social behavior [64]. An analysis of how informal spaces arise outside of formal planning is a new perspective for interpreting the transformation and reconfiguration of urban social spaces. In the process of the transformation and reconfiguration of urban social space, problems such as spatial inequality [65] and the declining cohesion of urban communities [66] have also arisen. How to solve these problems in the process of transformation to achieve sustainable urban development is an important research topic. Active urban renewal programs [55], a shared urban development concept [57], and a green policy that values care, the common, and the collective [36] are possible solutions.

It is not difficult to find that the existing studies focusing on the transformation and reconstruction of urban social space have diversified themes, both in terms of theoretical discussion and analysis of practical problems. However, a significant shortcoming is that the existing studies are mostly centered on corroborating or questioning the existing theories, and no new theories have been created. In addition, spatial inequalities and informal settlements (e.g., slums) that arise in the process of urban socio-spatial transformation and reconfiguration are serious impediments to the process of sustainable urban development and are topics of great concern in the SDGs [4]. Therefore, future research should focus more on how to optimize the socio-spatial structure of cities, to provide a more inclusive and safe living environment for all people, and to achieve sustainable urban socio-spatial development.

## 3. Social Composition of Urban Community Residents and Social Integration

The social space of the city is distributed with groups of residents of different age structures, different ethnicities, and different socio-economic conditions. As different social groups are integrated into the social macrostructure to varying degrees, they settle in neighborhoods with distinctive attribute characteristics and establish unique relationships within their communities [67]. This gives rise to a series of social problems, such as social exclusion and residential segregation, which pose a great challenge to the sustainable development of cities.

### 3.1. The Concept of Social Integration

Since the 1990s, "social integration" has gradually replaced "equality" as one of the central concepts in social policy practice and research on community composition [68]. The concept of "social integration" originated from European scholars' research on social exclusion and was originally proposed by Emile Durkheim, who defined social integration as the process by which individuals develop a collective consciousness based on the division of labor in society, thereby maintaining social order [69]. The concept of social integration has been used in many studies since then, but because of its complex, multidimensional, and dynamic nature, scholars have interpreted it differently in different contexts [70], resulting in definitions such as social inclusion [71] and social interaction [72]. However, these definitions have largely failed to cover multiple aspects of social integration [73]. Although there is no consensus on a formal definition of integration, most scholars consider social integration as a two-way process of mutual adaptation of different groups [74]. According

to Esser's classification, social integration includes four basic forms: structural integration, cultural integration, interactive integration, and identificational integration [75]. Structural integration means that people have equal access to public resources and social benefits such as education, the labor market, and health; cultural integration is the acquisition of knowledge and competencies about cultural aspects, such as language or general rules of behavior, which enable individuals to avoid cultural exclusion in society; friendships, partnerships, and other social aspects are interactive integration, meaning that people are included in the main networks and relationships of the local society; and finally, identificational integration is the sense of belonging to the society in which people live, that is, the emotional ties people have to the place or other local groups [76].

### *3.2. Theories of Social Integration*

In the study of social integration, social space is a factor that cannot be separated from consideration. On the one hand, social integration occurs in social space, which provides the place and context for social integration. The socio-spatial location of residents can reflect the extent to which individuals have access to the resources and opportunities provided by the social system and can also influence the process by which individuals form new social relationships and create new opportunities for themselves [67]. On the other hand, spatial integration is an extremely crucial part of the process of social integration [77] and is a central element in many theories of social integration.

The theory of spatial assimilation, which emerged from the theory of urban ecology combined with the theory of social status acquisition, has been central to the interpretation of social integration/exclusion in the American academy [31]. Most of the series of other theories that have emerged since then are the product of a critique or rethinking of this dominant theory [77]. The central idea of the spatial assimilation theory is that as social status increases, non-mainstream/disadvantaged groups will achieve spatial integration with the dominant race by moving into mainstream communities that offer better superiority in terms of public services and healthcare [78]. This theory views the transition from social exclusion to social integration of non-mainstream/disadvantaged groups as a natural process of resource acquisition and transformation, in which socio-economics is the most important consideration [77]. Subsequent scholars have made a series of modifications based on this theory, resulting in the contemporary spatial assimilation theory, which emphasizes the proximity and integration of non-mainstream/underprivileged groups with the majority rather than the unidimensional assimilation of non-mainstream/underprivileged groups by the majority [77,79], and the delayed spatial assimilation theory, which emphasizes that spatial assimilation may not continue to strengthen with generations [77,80] and so on.

The segmented assimilation theory, which builds on but differs from the spatial assimilation theory, focuses on the differences in the social integration of disadvantaged groups [31]. This theory does not deny the transformation of economic and social capital into residential space, as described in the spatial assimilation theory, but it further states that while the non-mainstream/disadvantaged try to enhance residential space, the mainstream group that holds power will isolate the non-mainstream/disadvantaged from themselves on the physical and social levels by influencing the allocation of space [81]. So, it can be seen that the spatial assimilation theory is concerned with the comparison between non-mainstream/underprivileged groups, that is, whether members of higher socio-economic status within groups live in communities with better conditions and more mainstream groups. The segmented assimilation theory, on the other hand, is more concerned with the comparison, or game, between non-mainstream/underprivileged groups and mainstream groups. In other words, while the spatial assimilation theory assumes that residential segregation between mainstream and non-mainstream/underprivileged groups will inevitably be gradually eliminated over time, the segmented assimilation theory assumes that for some groups, social integration will not occur due to the influence of mainstream social groups.

Some scholars have also questioned the assumption of the voluntary and inevitable integration of non-mainstream/underprivileged groups into the mainstream in the spatial assimilation theory. The residential preference theory argues that each group exhibits preferences based on culture and similarity and that residential preferences are closely related to their residential patterns, so the essence of social segregation may be the self-segregation of residents [82]. The pluralism theory argues for a pluralistic social and economic order because it sees assimilation theory as distinctly dominant group-centered and discriminatory against non-dominant/underprivileged groups [83]. In the exploration of assimilation and diversity theories, the group threat theory suggests that frequent interactions with out-group members threaten in-group members' identity, while the contact hypothesis suggests that interactions with out-group members reduce in-group members' prejudice [84]. Thus, the group threat theory supports pluralism, while the contact hypothesis supports assimilation (Table 3).

**Table 3.** Main theories of social integration.

| Theories of Social Integration | Description |
| --- | --- |
| **Spatial Assimilation Theory** | - This theory emerged from urban ecology and social status acquisition theories; it is fundamental in interpreting social integration/exclusion in the American context.<br>Central idea: as social status increases, non-mainstream/disadvantaged groups integrate spatially with the dominant race by moving to better-resourced communities. |
| **Segmented Assimilation Theory** | - This theory builds on the spatial assimilation theory but differs in focus, examining differences in social integration among disadvantaged groups. |
| **Pluralism Theory** | - The pluralism theory advocates for a pluralistic social and economic order, contrasting the dominant group-centered assimilation theory. |
| **Group Threat Theory** | - The group threat theory suggests frequent interactions with out-group members threaten in-group identity, supporting pluralism. |
| **Contact hypothesis** | The contact hypothesis proposes that interactions with out-group members reduce in-group members' prejudice, supporting assimilation. |

Overall, the theoretical paradigm on social integration is more diverse, with most theoretical explanations revolving around two explanatory lines: a socio-economic explanatory model and a cultural–ethnic explanatory model [77]. When analyzing specific issues, scholars combine different theoretical paradigms to produce realistic understandings.

### 3.3. Empirical Studies of Social Integration

There are a large number of empirical studies on social integration with more diverse research objects and concerns. In terms of research subjects, since those who are socially excluded are often non-mainstream/vulnerable groups in society [68], social integration research, which originated from social exclusion studies, also focuses on these groups, such as immigrants, ethnic minorities, the elderly, and the poor. One group that has received much attention is immigrants. Most social integration theories are developed in Western countries to understand and explain their immigrants [85], before being widely applied to social integration studies of other non-mainstream/vulnerable groups. Most residents may have multiple roles at the same time, and therefore, most studies consider multiple socio-economic characteristics of the study population simultaneously, such as occupation type and immigration status [86,87], age group and refugee status [88], etc (Table 4).

**Table 4.** Summary of Empirical Studies on Social Integration.

| Research Subjects | Measures of Integration | Factors Affecting Integration | Effects of Integration |
|---|---|---|---|
| Immigrants Ethnic Minorities The Elderly The Poor | - Multiple dimensions<br>- Economic, cultural, social, structural, and identity<br>- Education, law, social inclusion, and employment | - Socio-economic conditions<br>- Digital communication technologies<br>- Religion<br>- Education<br>- Host country attitudes | - Health<br>- Political participation<br>- Equal access to resources |

　　　　Studies on immigrant integration have focused on different scales in different countries, with developed Western countries focusing on international migration [86,89] and developing countries such as China focusing more on domestic migration [90–92] studies. However, in terms of research topics, different countries are concerned with the characteristics of immigrant integration, measures of immigrant integration [93,94], factors affecting immigrant integration [87,89,95], and effects arising from immigrant integration [91,96].

　　　　Scholars have different opinions on the system of indicators to measure the degree of immigrant integration. Hao Zhou included economic integration, cultural adaptation, social adaptation, structural integration, and identity in the measurement system [94], while Acikalin et al. measured immigrants' social integration from the perspectives of education, law, social inclusion, and employment [93].

　　　　In exploring the factors influencing immigrant integration, scholars have not only focused on socio-economic conditions [89], but also explored the effects of digital communication technologies [87], religion [97,98], education [95], and host country attitudes toward immigrants [99–101]. Studies of China's agricultural migrants have divided their integration into urban society into three stages, including circular migration, urban settlement, and urban integration, in which the economic status of urban and rural migrants, the expansion of social welfare, and the social interaction between agricultural migrants and urban residents are key factors influencing the transition between these stages [92]. For example, community identity and community participation can have a positive impact on the social integration of China's agricultural migrants [102]. An empirical study on the social integration of Syrian refugees in Germany shows that economic integration affects the social integration of low- and medium-educated refugees and that language proficiency, marital status, and social capital in the host country have positive effects on the social integration of all immigrant groups [89]. Research based on immigrants in deprived areas of the UK has shown that functional factors (educational qualifications, ability to speak English, and employment), time, and location have an impact on immigrants' social integration; while functional factors and time have a positive effect on different aspects of social integration, the location of residence is negatively associated with several aspects of immigrants' social integration [103]. The role that communication technologies can play in social integration has also received much attention in the context of the rapid development of digital communication technologies. A study of Chinese food delivery workers found that online platforms are both an opportunity and a barrier to integration for rural–urban migrants, as discourses promoting the digital economy and urbanization sometimes make them aware of the possibilities of social integration, but the digital order in online platforms can also reinforce to some extent the system of labor inequality between them and urban residents [99]. In addition to digital technology, religion is also an important factor in the integration of immigrants. Religion can make church members feel a sense of belonging in their new community by creating a strong sense of belonging internally, but religious differences can be a barrier to social integration for some immigrants [97]. Therefore, the role that religion can play in social integration at a holistic level is open to question. Other studies have focused on immigrant student populations, arguing that one of the functions of education is to strengthen the social integration or identity of students, especially immigrant students [104] and that cross-border participants in educational programs are more likely to integrate into the local culture and society [95]. However, the extent to which

education affects social integration is largely influenced by cultural distance. Cultural distance refers to differences in the native language, religion, life, values, etc., between immigrants and host countries [105]. A study of Hong Kong cross-border university students in mainland China shows that the greater the cultural distance, the more limited the role that education plays in social integration [95]. Host country attitudes toward immigrants are also a key factor in the social integration of immigrants, and studies of social integration in Europe have found that inclusion and good attitudes promote the social integration of all immigrant groups (EU, and non-EU immigrant groups) [99].

High levels of social integration can have numerous effects. Based on a study of Chinese agricultural migrants, it is found that high levels of social integration have a significant positive effect on physical health and that the degree of effect varies by gender, generation, and wage level, with women, recent generation migrants, and migrants at middle-income levels having a higher degree of health affected by social integration [91]. In addition, social integration plays a significant role in immigrants' political participation. Empirical studies show that the significant political participation differences between intra- and inter-provincial Chinese immigrants are mainly influenced by differences in the degree of social integration, with the latter having more difficulties in adapting to their new environment compared to the former, thereby undermining their opportunities to participate in local politics [96].

In addition to immigrant groups, non-mainstream/vulnerable groups such as ethnic minorities, elderly groups, and poor groups are also the focus of research on the topic of social integration, and the topics discussed in related studies are more similar to those in immigrant studies. Of these, studies on ethnic integration have focused on the United States [106,107]. Carrillo and Rothbaum's research shows that urban areas in the United States have experienced important changes in the distribution of race/ethnicity and that the spatial integration of races is influenced by individual characteristics such as education, income, and age, but spatial integration does not necessarily translate into better economic and social integration [108]. Unlike other members who experience migration in the home, older adults may be affected by social interactions, social participation, psychological acceptance, and self-identification which lead to greater social exclusion [69]. A study of elderly refugees in Turkey finds that health problems, language problems, and poverty greatly increase the difficulty of social integration of elderly refugees, with female elderly refugees being at a greater disadvantage [88]. Bezin and Moizeau's study of poor groups finds that urban minorities isolated in poor neighborhoods may create their own "culture of poverty" that deviates from mainstream socio-cultural behavior and keeps them in poverty [109]. A society with a high level of integration provides more equal access to opportunities and resources for different population groups, enabling non-mainstream/disadvantaged groups to live and work in the city, gain a foundation for living and building an appropriate social status, and sustainably accumulate the ability to integrate into society.

It is clear from the existing research that social integration is a topic of great interest in all countries, and the relevant theories are rich and diverse, including the study of non-mainstream/vulnerable groups such as immigrants, ethnic minorities, the elderly, and the poor, as well as considering the multiple social roles of the subjects.

Scholars have not only focused on the social integration characteristics of these non-mainstream/vulnerable groups but also explored in depth the factors that influence their integration process and the effects that integration can produce. However, as some scholars have argued, social integration is a problem that exists universally in social groups rather than a problem faced only by immigrants or disadvantaged groups, so the traditional dichotomous understanding of thinking is biased [85]. To translate these insights into effective policies, urban planners and policymakers can take a multifaceted approach. First and foremost, policies should acknowledge the multifaceted roles individuals play in society, considering factors such as occupation, immigration status, age, and refugee status simultaneously. By recognizing the diverse characteristics of these populations, urban policies can be tailored to meet their unique needs and challenges. Furthermore,

policy formulation should consider the variety of factors influencing social integration, ranging from socio-economic conditions and digital communication technologies to religion, education, and host country attitudes toward immigrants. Tailoring policies to address these diverse factors is essential for creating a more inclusive urban environment. For instance, policies can support education programs that enhance social integration and identity, especially for immigrant students, while being sensitive to the role of cultural differences in this process.

In conclusion, the extensive body of empirical research on social integration offers a comprehensive understanding of the intricacies involved in achieving social inclusiveness in urban areas. These findings serve as a solid foundation upon which urban policies can be built. By taking a holistic approach to social integration and adopting a multidimensional perspective, policymakers can effectively address the unique challenges faced by non-mainstream/vulnerable groups and promote the inclusiveness of urban social spaces, ultimately contributing to the sustainable development of urban communities.

## 4. Activity Space and Living Space of Urban Residents

With the rise of humanism, individual and microscopic studies have received increasing attention from scholars [110]. Human orientation has become the core of urban development in many countries/regions [111,112]. As seen in SDG 11, the livelihood of residents is a very important aspect of measuring urban sustainability; for example, ensuring that everyone has access to services such as housing, transportation, green public spaces, and inclusive and safe community environments are emphasized [4]. Therefore, when exploring the issue of urban sustainability, it is important to focus not only on macroscopic urban and community issues but also on the daily lives of residents. By focusing on the daily lives of residents, we can, on the one hand, see the overall and macroscopic urban socio-spatial development issues from the perspective of individual and micro-residents, and, on the other hand, discover the diverse needs of different social groups, effectively improve the quality of residents' lives, and thus promote the sustainable development of cities.

### 4.1. The Concept of Activity Space and Living Space

The study of residents' daily life can start with the residents' activity space and living space. The concepts of activity space and living space originate from behavioral geography and time geography. Activity space refers to the spatial area involved in the daily activities carried out by individuals to satisfy their needs and the movement between activities, while the latter refers to the spatial area involved in the series of activities that people maintain in their daily lives, which is a kind of home-centered activity space [110]. On this level, the activity space has a broader scope; in other words, the living space can be understood as a special kind of activity space. However, different scholars have different understandings. For example, Wang Li et al. deconstructed living spaces and argued that the constituents of living spaces at the level of the spatial system include urban resources, community types, and community resources, and those at the level of the socio-spatial system include social equity, spatial public, cultural equality, and value respect [113]. Regardless of the specific definition, the study of activity spaces and living spaces revolves around the daily life of residents to improve the quality of life of residents. Therefore, this Section combines the existing research on activity spaces and living spaces to examine them together.

### 4.2. Theories of Activity Space and Living Space

Doxiadis, a Greek scholar, first gave the connotation of urban living space science, and he proposed the concept of "habitat science" based on the study of urban life and living environment [114], which set off a wave of urban living space research and planning. Since then, many theories around activity space and living space have gradually emerged, among which two theoretical foundations, behavioral geography and temporal geography, occupy the core position.

Behavioral geography began with the behavioral revolution and is a new paradigm of human geography research emphasizing the microscopic perspective, behavioral processes, and the combination of subjectivity and objectivity [110]. Its core is to explore the interaction between human behavior and space in different geographical environments and pay attention to the thinking of "people" [115]. In the process of development, behavioral geography has absorbed closely related theories and methods such as time geography and the activity-based approach, integrating with neighboring disciplines (e.g., psychology) [116] and gradually diversifying, becoming an important theoretical paradigm in the study of residents' daily lives.

Time geography is a methodology proposed and developed by the Swedish geographer Hagerstrand and his Lund School in the late 1960s [117,118]. This methodology emphasizes the holistic nature of space–time, the constraints that individuals are subject to in space–time, etc. [110,118] and provides a human-based perspective and methodology by emphasizing the focus on individual activities from space–time [119]. Time geography and the activity-based approach are often combined as the core of space–time behavior research theory [120]. The theoretical framework of Ping Cai and Hagerstrand is the source of the theoretical foundation of the activity-based approach, with the former's research pointing to time and space as influential factors behind behavioral patterns and the latter providing a framework for activity-based decision making under spatio-temporal constraints [117,121,122]. Therefore, the activity-based approach provides a decision model of everyday life, focusing on and explaining differences in residents' behavior across different lifestyles [120]. In conclusion, theories based on the study of spatio-temporal behavior, which can portray the daily lives of individuals or groups in detail at the microlevel [123], are regarded as the core theories for the study of daily life [124] and have received widespread attention and application.

*4.3. Empirical Studies of Residents' Activity Space and Living Space*

Empirical studies on residential activity spaces and living spaces have been conducted from several aspects, and the topics explored include the identification of spatial boundaries of different residential activity spaces and living spaces [125,126], evaluation of spatial quality [127–129], characteristics of activity spaces and living spaces [130,131], and identification of influencing factors [132,133]. Some studies analyze all of the main daily activity characteristics of residents at the spatial level [131], whilst other studies summarize the corresponding spatial characteristics from specific behaviors [134,135], and the summary of characteristics is usually combined with the analysis of influencing factors [130,136]. In terms of research data, in addition to traditional data such as in-depth interview data [137] and activity diary survey data, new types of data such as GPS data [138,139] and cell phone signaling data [140,141] have been used. An in-depth interview is a detailed and targeted conversation conducted by the researcher or research team to understand the respondent's viewpoints, experiences, attitudes, feelings, etc. The questionnaire data are collected by asking a series of questions to the respondents and recording their responses. GPS data and mobile phone signaling data contain a large amount of geo-spatial information, both of which are different in accuracy but can dynamically record individual movement characteristics and vividly display the living and activity space of residents [141] (Table 5).

The quality of activity spaces and living spaces is one of the important indicators reflecting people's happiness in life and one of the key concerns in academic studies. Scholars have evaluated the quality of the living space of different people based on different data and from different dimensions [113,127–129,142]. Duan Zhaowen et al. collected data through a questionnaire survey to evaluate the quality of the living space for residents of public rental housing in terms of resource accessibility (educational resources, shopping facilities, medical resources, and leisure resources), satisfaction (housing conditions and community environment), and residents' sense of belonging [128]. Zhaozhong Li, on the other hand, evaluated the quality of the living space in Nanjing in terms of five dimensions,

including comfort, convenience, health and safety, and sociality, by combining multiple sources of data such as community attribute data, poi data, and environmental monitoring data [127]. Closely linked to the evaluation of living space quality is the analysis of residents' satisfaction with their living space and activity space. For example, using a questionnaire survey method to study workers' satisfaction with activity spaces in office areas, problems in office areas in terms of commercial support facilities and greening conditions can be identified [143]. The study of living space quality and satisfaction can provide a reference basis for urban planning, thus making it possible to effectively improve the well-being of residents.

**Table 5.** Summary of Empirical Studies on Residents' Activity Space and Living Space.

| Research Aspects | Description |
|---|---|
| **Research Topics** | - Spatial boundaries identification, quality evaluation, characteristics, and influencing factors. <br> - Daily activity characteristics, behavior-based spatial characteristics, and influencing factors analysis. |
| **Data Sources** | - Traditional data (in-depth interviews and activity diaries) or new data (GPS and cell phone signaling). |
| **Quality Evaluation** | - Multiple dimensions of quality (e.g., resource accessibility, satisfaction, and a sense of belonging). <br> - Evaluation based on different data sources and dimensions. |
| **Characteristics and Factors** | - Analysis of residents' activity spaces and living spaces, and identification of influencing factors. <br> - Socio-economic conditions, built environment, ICTs, and the impact of digitalization. |
| **Insights into Non-Mainstream Groups** | - Study of low-income clusters, children, older adults, refugees, and their daily activities. <br> - Mobility analysis, social relationships, social equity, and social inclusion issues. |
| **Urban Planning Implications** | - Basis for urban planning, improving residents' well-being, and promoting city sustainability. <br> - Recognition of diversity and challenges in activity and living space design. |

The characteristics of residents' activity spaces and living spaces and the factors which influence them are central research topics that have received much scholarly attention. For example, a Nanjing-specific study based on activity diary survey data found that low-income participants who are highly dependent on the central city for their daily activities (especially work) are disadvantaged in terms of access to the central city, compared to non-low-income people who have more diverse activity spaces and can more fully utilize various types of urban areas; the factors influencing this difference include activity characteristics and the built environment, in addition to income [130]. A survey study of community residents in the Beijing Economic and Technological Development Area based on questionnaire data points out that distance, business preferences, requirements for quality of service facilities, and changes in residents' demand due to improved transportation modes, together with the spatial distribution of market supply and the relative shortage of government supply, shape the daily activity space of community residents [131]. In addition to these socio-economic conditions and built environment factors, the rapid development of ICTs has led scholars to gradually focus their attention on the impact of ICTs [128,132,144]. Nowadays, more and more individuals can access the Internet, and people's daily life has changed considerably with the emergence of some new characteristics. These features can be summarized as fragmentation, multitasking, and space–time substitution, where fragmentation refers to the original activity being broken up into multiple sub-activities then widely and discontinuously distributed in time and space, multitasking refers to the phenomenon of people carrying out multiple activities at the same time with the help of ICTs, and space–time substitution refers to the substitution of online activities for offline activities [128]. A study of female residents in Beijing's urban villages finds that young women not in a marital relationship have a high level of education and are in a better economic situation than the majority of those who engage in Internet activities, and ICTs are not an influential factor in their choice of living space but can weaken their time and space constraints on leisure activities [144]. These studies of ICTs help urban planners to respond positively to the changes that digitalization brings to the lives of residents.

The study of living spaces and activity spaces provides important insights into the problems of non-mainstream/vulnerable groups such as low-income groups, the elderly, and immigrants. Studies of low-income population clusters, such as urban villages, have found that the daily activities of the population have certain regularities [136] but also exhibit individualized and diverse characteristics [145], which are the result of multiple factors such as the macro-environment and their characteristics. Some scholars have also focused on children in urban villages. Through in-depth interviews, they have found that children's outdoor activity space is mainly concentrated in places with spacious grounds and high store density, but there are differences between children of different ages [146]. A study of older adults based on household travel survey data found that there are two main activity patterns of older adults, the recreation–shopping-oriented (RS-oriented) pattern and the schooling–drop-off/pick-up-oriented (SDP-oriented) pattern, with RS-oriented older adults facing spatial constraints and SDP-oriented older adults being more likely to experience time constraints when making daily trips [147]. Mobility in living spaces is a representation of an individual's ability to move in the context of daily life [148]. A study on the mobility of elderly people's living space divides the influencing factors into non-physical environment elements and physical environment elements: non-physical environment elements include home environment and social culture; physical environment elements are divided into objective environment elements and subjective perception elements, where objective environment elements refer to physical indicators such as density, diversity, and street connectivity, while subjective perception elements refer to perception indicators such as safety, accessibility, comfort, and pleasure [149]. A study of refugees based on in-depth interviews found that daily life in neighborhood spaces provides opportunities for refugees to develop and maintain social relationships, but that integration with residents is difficult because some potential social spaces are legally or economically inaccessible [150]. It can be seen that activity spaces and living spaces are effective entry points to analyze the issues of social equity and social inclusion faced by non-mainstream/vulnerable groups.

The study of living spaces and activity spaces starts from the micro- and individual level, studying the daily life of residents to summarize the characteristics of residents' daily lives and discover the problems and influencing factors. It can both summarize the daily life patterns of mainstream resident groups and discover the social injustice problems faced by non-mainstream/underprivileged groups. These findings emphasize the importance of optimizing resource allocation to ensure equitable access to essential services, enhancing housing and community design to create safe and comfortable environments, promoting digital inclusion for all residents, addressing spatial inequalities through flexible zoning and mixed-use development, actively engaging the community in the planning process, adopting an adaptive approach to accommodate evolving trends, and, above all, adopting a holistic and inclusive approach to urban planning. In conclusion, empirical studies on activity and living spaces offer a wealth of information that can be invaluable for urban planning and policymaking. By incorporating these perspectives, policymakers can consider various viewpoints and communities while developing urban policies that enhance the quality of life for residents and facilitate sustainable urban growth.

## 5. Urban Planning and the Sustainable Development of Urban Social Space

Cities are protagonists in the sustainable development of society, and urban planning offers a variety of solutions to the complex problems and challenges of sustainable urban development. In an era of public skepticism toward science, sustainability science offers a way to increase the application of science in planning and policy [151,152]. Urban planning can develop solutions to a range of urban problems at the national, city, and community levels and can also involve residents through bottom-up planning. It is an important tool for achieving sustainable development. Therefore, this Section will review the urban planning concepts and practices that have received much attention in the existing studies from three dimensions: city, community, and individual.

### 5.1. Frontier Concepts of Urban Planning in the 21st Century

The idea of planning at the city level plays a guiding role in planning at the community and individual levels. Modern urban planning emerged over 100 years ago and has been divided into three stages: the first stage was concerned with scientific rationality and attempted to solve the problem of material planning, represented by the Athens Charter in the 1930s; the second stage was advocacy planning, which shifted the focus to the equity of the planning process, with special attention paid to disadvantaged groups, and attempted to solve the problem of sociological planning; the third stage belonged to collaborative planning, concerned with the collective rationality of the planning process, and emphasized the concept of sustainable development, and the theories of this stage included New Urbanism and Smart Growth, which emerged after the 1980s and 1990s [153]. In the 21st century, frontier urban planning concepts can be divided into six categories: sustainable development, information and digital technology, urban social and community life improvement, political or economic, conservation and renewal as the core, and other planning concepts (such as Infrastructural Urbanism, Stereoscopic Urbanism, etc.), among which, sustainable development and information and communication technology are the core drivers for creating new urban forms in the 21st century [154]. As can be seen, sustainable development has become a central goal in current urban planning. Based on cutting-edge urban planning concepts, new urban concepts such as sustainable cities [155–161], resilient cities [162–166], smart cities [162,167–169], and healthy cities [170–172] have been derived in academic and political circles. Sustainability is a multidimensional concept, and sustainable cities focus not only on sustainability in the physical spatial realm of cities but also in the socio-spatial realm of cities, involving multiple dimensions such as environmental, economic, social, and cultural dimensions [173]. The term resilience originated in physics to describe the ability of an object to recover from deformation under the action of external forces [166]. After being introduced to urban planning, it was defined as the ability of individuals, communities, institutions, etc., within a city to survive, adapt, and grow after experiencing shocks [174]. The definition of smart cities is not unique and has understandings based on technology, knowledge, and governance perspectives, with the technology-centric perspective dominating [175,176]. Healthy cities focus on healthy people, a healthy environment, and a healthy society and hope to enhance the health of residents in urban societies through the implementation of planning programs such as urban health projects [170]. These new urban planning concepts are not only widely discussed in academic circles but also applied to actual urban planning processes in different countries [165,174,177,178]. For example, Oswiecim realizes the sustainable development of towns by enriching their functions, designing town spaces based on the concept of balance, and the main measures include the provision of jobs and housing to meet the basic needs of Oswiecim's residents, increasing the accessibility of local sports and recreational venues, good safety monitoring, taking care of the needs of each social group, and encouraging social participation [177]. Japan focuses on public participatory resilience building with the community as the basic unit, and about 30% or more of Kyoto's resilience strategy emphasizes the importance of community and public participation in reducing the risk of future shocks. The Kyoto government encourages citizens to participate in improving the safety and security of their neighborhoods and cities and builds communication channels between the government and citizens in the community through smart programs to maintain and strengthen the assets of urban facilities while also forming a good government–society partnership [165]. The construction of smart cities has also received attention from many countries, and smart governance, smart people's livelihood, and solving problems such as traffic congestion, environmental pollution, and resource constraints are the development goals of smart cities. Shanghai is one of the earliest pilot cities in China to build a smart city, and Shanghai has made great progress in information infrastructure construction, information perception and intelligent application, a new generation of the information technology industry, information security assurance, etc. [178]. The sixth phase of the European Healthy Cities Program supports cities to strengthen linkages, bring key stakeholders together for health

and well-being, and use leadership, innovation, and change to increase the potential to address local public health challenges [171]. It can be seen that many countries are applying these concepts to urban development planning to create more livable, sustainable, safe, and healthy urban environments. However, there are limitations to the application of these urban planning concepts, such as the need for large-scale investment in the construction of resilient cities; the construction of smart cities, which may bring about issues of privacy and security, digital divide, and technological dependency; the construction of healthy cities, which may also lead to issues of health inequality; and the planning of sustainable cities, which requires weighing conflicting interests, such as those of the environment, society, and the economy. Therefore, by applying these cutting-edge concepts, urban planners and governments need to develop integrated policies that encourage innovative, diverse solutions and ensure equitable, inclusive, and sustainable urban development.

### 5.2. Concepts and Solutions for Urban Community Planning

In addition to city-level planning concepts and programs, there are also many community-level planning concepts and programs that aim to improve social integration and enhance the well-being of residents, such as the "15-min Community Life Circles" planning and urban housing projects. The concept of life circles originated in Asian countries and cities and was first introduced in Japan in 1965 in the Second National Comprehensive Development Plan as a planning strategy to promote social equity through the rational arrangement of basic services and facilities [179]. The concept was proposed and combined with the later concept of 15-minute cities as a new solution for community-level planning in many countries [180–182]. The concept of 15-minute cities was proposed by Carlos Moreno in 2016 and defines a highly flexible urban model [150,183] with several characteristics such as proximity, density, diversity, mixed-use, modularity, and adaptability [184]. The concept was introduced and sparked many controversies. Firstly, there were disagreements on the definition of time as 20 min, 30 min, or even longer [181], and secondly, the concept followed the philosophy of physical determinism, setting goals but not specifying how to achieve them [184]. However, it is undeniable that this planning concept focuses on the basic needs of residents and communication and interaction and is extremely humanistic. Empirical studies have shown that such planning solutions can help increase urban resilience, especially in the context of climate change, pandemics, and other issues, while also reducing inequalities between different parts of the city [185]. In addition to community-level life circle planning, housing projects are also planning programs which have been adopted by most countries to promote social integration. In Chile, housing programs have been implemented to promote social integration [186,187]. In Albania, housing production is seen as key to promote the integration of various social groups into the urban development process [188], and in China, the government has implemented policies to relocate poor people and thus improve their living conditions [189]. A study based on China's affordable housing program finds that the implementation of the program contributes to the integration of migrants into urban society [190]. In conclusion, positive planning at the community level follows a humanistic ideology and aims to enhance community inclusiveness and promote sustainable community development.

### 5.3. Public Participation in Urban Planning

When urban planning focuses its target on individuals, the conversation revolves around public participation, that is, bottom-up planning. The controversy over whether public participation can contribute to urban planning has existed for many years, with some scholars arguing that public participation does not play a great role [191] and others pointing out that public participation is not only an important manifestation of government democracy but also an important initiative to enhance the harmonious relationship between the government, the planning department, and the people [192]. Some scholars have also shown that the combination of bottom-up and top-down planning approaches helps to match the needs of residents with the urban socio-spatial development vision of decision

makers [193]. For example, the UK spatial planning guidance recognizes the importance of involving stakeholders in the assessment process at an early stage of the planning process [194]. Since the 1970s, European countries have adopted public hearings and discussion groups to promote public participation in urban planning [195]. In Chicago, Illinois, USA, São Paulo, Brazil, and Delhi, India, public participation is a key tool used in the development of their planning documents and can play a role in achieving social equity and justice [196]. In short, resident participation in planning can be a complement to top-down planning, while at the same time giving individuals a way to express their wishes and needs, thus promoting social equity to a certain extent. But how to combine the two properly is a question that needs to be explored.

Urban planning is the field of study that focuses most explicitly and specifically on urban areas and is able to be applied at multiple levels—city, community, and individual—to address a variety of issues that arise in urban development. However, there is controversy about the role that urban planning can play, and in the United States, for example, the planning department has very limited powers, making it difficult for urban planning programs to achieve the appropriate goals [197]. A study based on the city of Changchun, China, also finds that urban planning, although it plays a role in guiding the employment activities of residents, is unable to play other roles originally planned. This shows that although urban planning can provide many good visions, cities, communities, and residents do not always develop as planners expect, and this is a problem that needs more attention in future urban planning studies.

## 6. Conclusions and Future Research Directions

### 6.1. Conclusions

The research on urban socio-spatial and sustainable development is diversified, with rich studies at the macro- (city), meso- (community), and micro- (residents) levels, and urban planning concepts and practical solutions at all three levels have been fully explored (as shown in Figure 2).

At the urban level, the transformation and restructuring of urban social space is one of the most significant changes in the process of social transformation. Theoretical perspectives such as "socio-spatial" dialectic, anti-racist feminist–materialist urban theory, and neoliberalism have been introduced into the research in this field. The emergence of homogeneous to heterogeneous spatial structures, gentrification, and other characteristics of urban socio-spatial transformation are identified. The influencing factors such as administrative forces, market forces, social forces, and ICTs and their mechanisms of action are also identified. However, it is not difficult to find that the existing studies mostly corroborate or criticize the existing theories when focusing on the reconfiguration of urban social space, without establishing new theories. In addition, the solutions to the social problems emerging from the transformation and reconfiguration of urban social space have not been explored enough.

At the community level, communities are composed of groups of residents with different age structures, different ethnicities, and different socio-economic conditions. The integration between these different resident groups contributes to the sustainable development of the community. A large number of theories such as the spatial assimilation theory, the segmented assimilation theory, and the diversity theory have been applied to the study of social integration issues. The research subjects are mostly non-mainstream/vulnerable groups, such as immigrants, ethnic minorities, the elderly, and the poor, and some studies consider the multiple social roles of the research subjects at the same time. In the process of social integration, many factors such as socio-economic conditions, education, religion, culture, ICTs, etc., have facilitated or inhibited the integration process, respectively. However, existing studies have largely explored social integration based on the dichotomy of mainstream groups versus non-mainstream/vulnerable groups, ignoring the fact that social integration is a problem that exists universally among social groups.

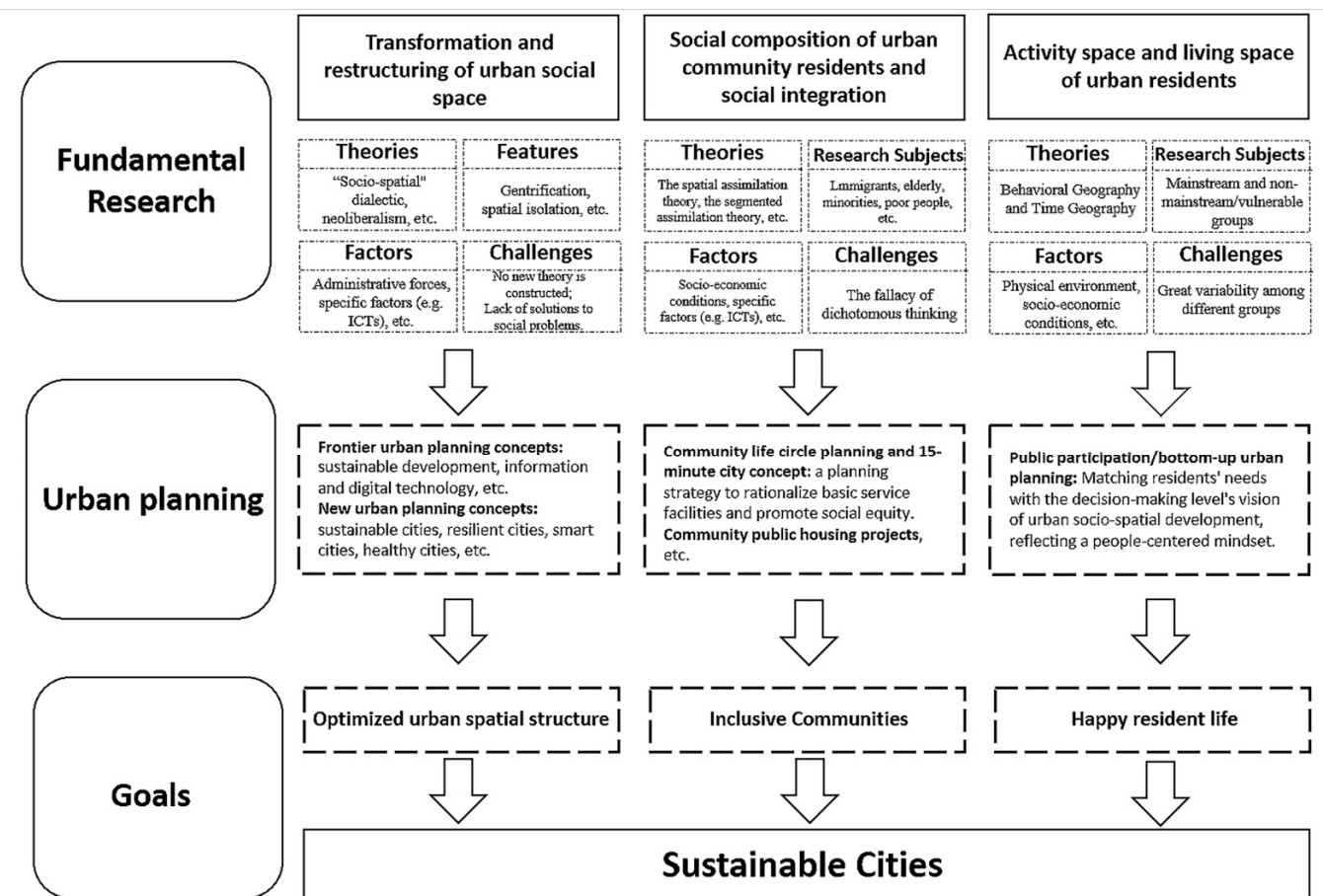

**Figure 2.** Summary of the combining of urban space and sustainable development research.

At the resident level, activity spaces and living spaces provide individual, microscopic perspectives for analyzing and improving residents' problems. Most of the relevant studies are based on behavioral and temporal geography theories. Research from residents' activity spaces and living spaces can focus on both mainstream groups and non-mainstream/vulnerable groups. Some studies analyze all of the major daily activities of residents from the spatial level, while other studies analyze the corresponding activity spaces from specific behaviors, and the studies cover various aspects such as evaluation of the quality of activity spaces and living spaces, analysis of characteristics, and identification of influencing factors. It can be seen that the research at the resident level starts from the microscopic level, from the individual needs, based on which the corresponding planning can effectively improve the happiness of the residents. However, the design and implementation of the corresponding urban planning can be very difficult due to the huge differences among resident groups.

Basic research at the city, community, and resident levels is the reference point for urban planning, and there are many new concepts and practical solutions in the field of urban planning to address issues at these three levels. At the city level, frontier urban planning concepts revolve around sustainability and information and communication technologies, resulting in new urban concepts and urban planning projects such as sustainable, resilient, smart, and healthy cities. At the community level, community life circle planning combined with the 15-minute city concept has emerged as a new solution to community problems. Urban housing projects are also being implemented in several countries to promote social inclusion. At the resident level, the issue of public participation has received much attention, and many countries have made public participation an important part of their planning. Urban planning is an important tool for solving urban problems. However, the question of how to make urban planning work as intended must be addressed.

*6.2. Future Research Directions*

Despite the extensive research conducted in the field of urban social space and sustainable development, there are still some shortcomings. Moving forward, future research and practice should aim to better achieve the goal of sustainable urban development.

At the city level, new theories of urban socio-spatial reconfiguration need to be actively explored in order to adapt to the ever-changing urban social structure. These new theories should be able to better address the transition from homogeneous to heterogeneous urban structures and offer more practical solutions to the social problems caused by urban transformation.

At the community level, future research should continue to focus on interactions and cooperation among residents, including those from different age groups, ethnicities, and socio-economic backgrounds. Additionally, research should explore strategies for building more inclusive communities. Focusing on community governance and resident participation, it is important to explore and utilize the role of digital technology in community interaction as much as possible. These future research directions will contribute to a more comprehensive understanding of sustainable development issues at the community level and provide more focused recommendations for community planning and policymaking.

At the resident level, it is important to prioritize the needs of vulnerable groups, such as the elderly and the poor, in order to ensure that all social groups have access to the resources and opportunities offered by the city. Programs will be sought to enhance the well-being of all residents, with a focus on their living space and mobility. In addition, the study can explore the relationship between sustainable lifestyles and residents' behaviors. How to encourage and support residents to adopt more environmentally friendly lifestyles, including energy conservation, waste reduction, and sustainable transportation, is an area of research closely related to sustainable development.

At the planning level, it is important to continue exploring ways in which urban planning can make a real difference. This may require broader public participation to ensure that the planning process reflects the needs and perspectives of diverse social groups.

In addition, sustainable urban development requires proper coordination among the social, environmental, and economic dimensions. Therefore, future research needs to pay more attention to the interactions among these three pillars in order to find more effective ways to promote coordination and synergies among them, ultimately achieving sustainable development in cities at all levels.

In conclusion, through ongoing in-depth research conducted at the city, community, and resident levels, and by applying this research to urban planning practices, we can expect to achieve a more optimized urban spatial structure, more inclusive communities, and happier residents' lives. This, in turn, will promote sustainable urban development. This will require continuous innovation and interdisciplinary collaboration to address the complex challenges of urban socio-spatial and sustainable development.

**Author Contributions:** Conceptualization, J.F. and H.H.; methodology, J.F. and H.H.; software, H.H.; validation, J.F. and H.H.; formal analysis, J.F.; investigation, H.H.; resources, J.F.; data curation, H.H.; writing—original draft preparation, J.F. and H.H.; writing—review and editing, J.F. and H.H.; visualization, H.H.; supervision, J.F.; project administration, J.F.; funding acquisition, J.F. All authors have read and agreed to the published version of the manuscript.

**Funding:** Supported by the National Natural Science Foundation of China (42171194).

**Institutional Review Board Statement:** Not applicable.

**Informed Consent Statement:** Not applicable.

**Data Availability Statement:** No new data were created or analyzed in this study. Data sharing is not applicable to this article.

**Conflicts of Interest:** The authors declare no conflict of interest. The funders had no role in the design of the study; in the collection, analyses, or interpretation of data; in the writing of the manuscript, or in the decision to publish the results.

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
