# Peer review of "Review of Research on Urban Social Space and Sustainable Development"

_sustainability, doi:10.3390/su152216130_

Round 1

Reviewer 1 Report

Comments and Suggestions for Authors

The current review paper is content-rich, which can provide readers with valuable structured knowledge of social space and sustainable urban development. However, several issues must be addressed before I can recommend the paper for publication.

-        First, the Abstract is highly ambiguous. The statement indicating the paper’s objective is excessively lengthy, with seven lines. The review’s findings and their implications are not displayed.

-        Sustainable development is a vast spectrum encompassing many aspects, so it is necessary to define the conceptual boundary of sustainable development in this review.

-        The structure of the whole manuscript needs to be clarified at the end of the Introduction to present the logical flow of the review and guide readers.

-        Line 36: Remove “However”

-        The statement in Lines 50-52 is confusing. What are “the global cities in 50 the international”?

-        Lines 281-342 should be broken down into smaller paragraphs for clarity and readability.

-        The current review is missing a crucial research agenda for urban social space and sustainable development. The sustainable development of cities cannot be achieved by studying only the components within the geographical area of cities, as it is interlinked with non-urban ecosystems. The resources and services provided by natural ecosystems are the sine qua non of urban existence. Therefore, urban society must be involved in the activities that help reduce climate change, biodiversity loss, and environmental degradation (e.g., carbon trade, conservation donation, unsustainable consumption, etc.). To realize such a goal, the eco-surplus culture should be built to replace the omnipresent eco-deficit culture among urban residents (https://www.nature.com/articles/s41599-022-01441-9). Exploring how urban social space can help contribute to the development of eco-surplus culture among urban residents is a pivotal research agenda for achieving sustainable development, besides solving social issues in urban areas (https://reunido.uniovi.es/index.php/EBL/article/view/15872).

Comments on the Quality of English Language

Minor editing of English language required

Author Response

Quality of English Language  (x) Minor editing of English language required

Thank you for your suggestion. I have made revisions to the English language usage and have indicated all modifications through highlighting.

  1.  First, the Abstract is highly ambiguous. The statement indicating the paper’s objective is excessively lengthy, with seven lines. The review’s findings and their implications are not displayed.

Thank you very much for the advice, it's really something I need to be aware of. I have made revisions to the abstract accordingly. The updated abstract now includes a comprehensive overview of the problem being addressed, the findings of the research paper, and the potential implications of the study. Additionally, the revised abstract explicitly highlights that the article is a literature review. The modified abstract is presented below:

“Urban social space and sustainable urban development are both prominent areas of research in urban studies. The development of a city is closely tied to the development of its social space. The level of sustainable development in a city can be assessed by examining the evolution of its urban social space. Therefore, the relationship between the two is highly interconnected. However, the social dimension of sustainable development has always received the least attention compared to the economy and the environment. Therefore, this paper examines urban transformation and urban spatial structure, social composition and integration of residents in urban communities, the activity space and living space of urban residents, and the social integration of urban residents, activity space and living space, and urban planning, this paper provides a comprehensive literature review of research on urban social space and sustainable development. On the one hand, this literature review thoroughly examines the correlation between urban social space and sustainable urban development. On the other hand, it broadens the perspectives of urban research, highlights the key role of social dimensions in sustainable urban development, and helps to draw the attention of academics to this topic. In addition, this literature review may provide policymakers with more comprehensive information on urban social space and sustainable development.”

  1. Sustainable development is a vast spectrum encompassing many aspects, so it is necessary to define the conceptual boundary of sustainable development in this review.

Thank you very much for the suggestion, it is really something I need to add. In response, I have incorporated the concept of sustainable development into the review by consulting the relevant literature. Specifically, I have made specific additions to the introduction section. These additions are as follows:

“Social sustainability focuses on social well-being, emphasizing social equity, social cohesion, etc., and focusing on a variety of topics such as the basic needs of people's daily lives (e.g., housing, education, transportation, etc.), cultural integration, and other topics, with an emphasis on the goals of addressing social inequality and improving the well-being of residents. [7,8,9]”

“Overall, the study of urban socio-spatial and sustainable development focuses on both socio-spatial elements (e.g., urban socio-spatial structure, living space, activity space) and social sustainability (e.g., community integration, improvement of residents' well-being, reduction of social inequality, etc.).”

  1.  The structure of the whole manuscript needs to be clarified at the end of the Introduction to present the logical flow of the review and guide readers.

I greatly appreciate your suggestion, and I apologize for initially missing that part. In response, I have made revisions to the structure of the paper by incorporating the following additions at the conclusion of the introduction.

       “To furnish the readers with a coherent roadmap, this paper is meticulously structured as follows: Chapter 1 undertakes the task of not only introducing the intrinsic and intricate connection between urban social space and sustainable development but also elucidates the imperative significance of delving into this symbiotic relationship. Chapters 2 through 5 meticulously navigate the trajectory of research progress, traversing a spectrum that spans from the macroscopic to the microscopic, from theoretical frameworks to empirical substantiation, and from academic inquiry to practical implementation. These chapters delve into four distinct facets of urban social space and sustainable development, thereby providing a comprehensive perspective: first, the transformation and restructuring of urban social space; second, the social composition of urban community residents and social integration; third, activity space and living space of urban residents; and fourth, an in-depth review of urban planning in the context of sustainable development. The last chapter is the conclusion and discussion of this paper.”

  1. Line 36: Remove “However”

Thank you for the suggestion, there really is no transitive relationship in this place, and “However” should not be used, I have removed it.

  1. The statement in Lines 50-52 is confusing. What are “the global cities in 50 the international”?

Thank you very much for your suggestion. The phrase "the global cities in the international" in my post was intended to describe the cosmopolitan cities on an international scale. My original description was indeed confusing. "I have changed it to 'global cities'."

  1.  Lines 281-342 should be broken down into smaller paragraphs for clarity and readability.

      Thank you very much for your suggestion, lines 281-442 are indeed too long. I have made the section more readable by logically subdividing lines 281-243. The specific subparagraphs are: "the system of indicators to measure the degree of immigrant integration," "the factors influencing immigrant integration," and "the effects of immigrant integration." Each of these topics is discussed in separate paragraphs.

  1. The current review is missing a crucial research agenda for urban social space and sustainable development. The sustainable development of cities cannot be achieved by studying only the components within the geographical area of cities, as it is interlinked with non-urban ecosystems. The resources and services provided by natural ecosystems are the sine qua non of urban existence. Therefore, urban society must be involved in the activities that help reduce climate change, biodiversity loss, and environmental degradation (e.g., carbon trade, conservation donation, unsustainable consumption, etc.). To realize such a goal, the eco-surplus culture should be built to replace the omnipresent eco-deficit culture among urban residents (https://www.nature.com/articles/s41599-022-01441-9). Exploring how urban social space can help contribute to the development of eco-surplus culture among urban residents is a pivotal research agenda for achieving sustainable development, besides solving social issues in urban areas (https://reunido.uniovi.es/index.php/EBL/article/view/15872).

Thank you very much for your suggestions. I wholeheartedly agree with you that ecological sustainability is indeed the foundation of social sustainability and is crucial for achieving sustainable urban development. The three pillars of the sustainability discourse - economic, environmental, and social - are inextricably linked. Both economic sustainability and ecological sustainability are crucial for social sustainability. However, my article focuses on the sustainability of social spaces and is primarily concerned with social issues such as social equity and the well-being of residents. If I were to dedicate more time to discussing environmental sustainability, it would be necessary to also address economic sustainability in order to maintain logical coherence. However, this would result in a longer and less focused essay. Thanks again for your suggestions.

Reviewer 2 Report

Comments and Suggestions for Authors

The comments are as follows:

1.L17-18:Too many keywords; keywords should highlight the topic and focus of the paper as much as possible

2.L22:While the article discusses the concept of "sustainable development," it might be beneficial to provide a brief definition or explanation of what sustainable development entails. This could help readers who may not be familiar with the term.

3.L35:Clarify what is meant by "urban social sustainability." While it is mentioned as the least defined dimension, a brief explanation or examples of what aspects it encompasses would be helpful for readers.

4.L269:This somewhat overlong section provides an extensive overview of empirical studies on social integration, covering various groups like immigrants, ethnic minorities, the elderly, and the poor. To enhance clarity, you might organize this information into subsections or tables, categorizing the studies by the group studied and the key findings.

5.L290:The initial letter M is capitalized.

6.Given that social integration is crucial for sustainable urban development, consider including a brief discussion on policy implications. How can the findings from empirical studies inform urban policies aimed at enhancing social integration and inclusiveness in urban social spaces?

7.L438:Same as mentioned above. The section mentions various aspects of empirical studies on activity and living spaces. To enhance clarity, you might organize this information into subsections or tables, categorizing the studies by the topics explored (e.g., spatial boundaries, quality evaluation, characteristics, influencing factors).

8.Discuss how the findings from empirical studies on activity and living spaces can inform urban planning and policy decisions aimed at improving residents' well-being and promoting sustainable development. What practical recommendations can be drawn from this research?

9.L446-449:When discussing empirical studies, provide insights into the types of data sources used (e.g., GPS data, cell phone signaling data, interviews, surveys). This information can help readers understand the methodologies employed.

10.L541:Incorporate specific case studies or examples of how these cutting-edge urban planning concepts have been applied to real-world urban development. This can make these concepts more concrete and relevant to the reader. At the same time, the validity and challenges associated with these cutting-edge concepts can be critically assessed. Should the limitations or criticisms of these approaches be discussed?

Comments on the Quality of English Language

Moderate editing of English language required

Author Response

1.L17-18:Too many keywords; keywords should highlight the topic and focus of the paper as much as possible

       Thank you very much for your suggestion, I have modified the keywords. The modifications are as follows: deletion of the keywords "urban transformation; urban spatial structure; social integration; activity space; living space; urban planning" and addition of the keyword Literature review. The keywords "urban transformation; urban spatial structure; social integration; activity space; living space; urban planning" have been deleted, and the keyword "Literature review" has been added. It avoids the problem of too many keywords, and at the same time, it can make the readers clear that this paper is a literature review.

2.L22:While the article discusses the concept of "sustainable development," it might be beneficial to provide a brief definition or explanation of what sustainable development entails. This could help readers who may not be familiar with the term.

       Thank you very much for pointing out my problem. As per your suggestion, I have explained the term "sustainable development" in the article, and the added content is listed below:

“Simply put, sustainability is a concept that addresses questions and concerns regarding what defines a quality life in a specific location and era [2]. This involves evaluating the utilization and preservation of available resources to support the future needs of both human populations and the environment. The implementation of sustainable practices within a system is crucial to minimize negative impacts on both society and the natural world.”

       In addition, I have further explained social sustainability, which is the main focus of this paper, as stated in the third review comment, so thank you again for such a pertinent suggestion.

3.L35:Clarify what is meant by "urban social sustainability." While it is mentioned as the least defined dimension, a brief explanation or examples of what aspects it encompasses would be helpful for readers.

       Thank you very much for your suggestion, it is indeed something I need to add. After referring to the relevant literature, I have defined the concept of urban social sustainability in the review, briefly explaining what it consists of. The additions are in the introduction and are listed below:

       “Social sustainability focuses on social well-being, emphasizing social equity, social cohesion, etc., and focusing on a variety of topics such as the basic needs of people's daily lives (e.g., housing, education, transportation, etc.), cultural integration, and other topics, with an emphasis on the goals of addressing social inequality and improving the well-being of residents. [7,8,9]”

“Overall, the study of urban socio-spatial and sustainable development focuses on both socio-spatial elements (e.g., urban socio-spatial structure, living space, activity space) and social sustainability (e.g., community integration, improvement of residents' well-being, reduction of social inequality, etc.).”

4.L269:This somewhat overlong section provides an extensive overview of empirical studies on social integration, covering various groups like immigrants, ethnic minorities, the elderly, and the poor. To enhance clarity, you might organize this information into subsections or tables, categorizing the studies by the group studied and the key findings.

       Thank you very much for your advice. This section is really lengthy. I've divided this section into logical sections based on your suggestions, making it more clear and readable. The specific segmentation method is: "the system of indicators to measure the degree of immigrant integration", "the factors influencing immigrant integration", and "the effects of immigrant integration" are each one paragraph.

       At the same time, I have also added a table in this section to show the overview of existing empirical studies in a more concise and clearer way for the readers to read, the details of the table are as follows:

Table 4 Summary of Empirical Studies on Social Integration

Research Subjects

Measures of Integration

Factors Affecting Integration

Effects of Integration

Immigrants

Ethnic Minorities

The Elderly

The Poor

- Multiple Dimensions

- Economic, cultural, social, structural, identity

- Education, law, social inclusion, employment

- Socioeconomic conditions

- Digital communication technologies

- Religion

- Education

- Host country attitudes

- Health

- Political Participation

- Equal access to resources

5.L290:The initial letter M is capitalized.

       Thank you very much for reading my manuscript in such detail and for your suggestions. I re-examined line 290, which reads "Measured immigrants' social integration from the perspectives of education, law, social...", and I assume you are referring to the need to capitalize the M as the first letter of the word "measured". But a more complete sentence would be "..., while Acikalin et al. measured immigrants' social integration from the perspectives of education, law, social inclusion, and employment [81]. " As you can see from the complete sentence, the word "measured" should be a subjunctive verb in this case, so I don't need to capitalize the initial letter M. Thank you very much for your suggestion, and if I misunderstood, please point it out again, thank you!

6.Given that social integration is crucial for sustainable urban development, consider including a brief discussion on policy implications. How can the findings from empirical studies inform urban policies aimed at enhancing social integration and inclusiveness in urban social spaces?

       Thank you very much for your suggestion, it is indeed important to discuss how empirical research related to social integration can support urban policies aimed at enhancing social integration and inclusiveness in urban social spaces. Therefore, I have added this section to the article as follows:

       “To translate these insights into effective policies, urban planners and policymakers can take a multifaceted approach. First and foremost, policies should acknowledge the multifaceted roles individuals play in society, considering factors such as occupation, immigration status, age, and refugee status simultaneously. By recognizing the diverse characteristics of these populations, urban policies can be tailored to meet their unique needs and challenges. Furthermore, policy formulation should consider the variety of factors influencing social integration, ranging from socioeconomic conditions and digital communication technologies to religion, education, and host country attitudes towards immigrants. Tailoring policies to address these diverse factors is essential for creating a more inclusive urban environment. For instance, policies can support education programs that enhance social integration and identity, especially for immigrant students, while being sensitive to the role of cultural differences in this process.

In conclusion, the extensive body of empirical research on social integration offers a comprehensive understanding of the intricacies involved in achieving social inclusiveness in urban areas. These findings serve as a solid foundation upon which urban policies can be built. By taking a holistic approach to social integration and adopting a multidimensional perspective, policymakers can effectively address the unique challenges faced by non-mainstream/vulnerable groups and promote the inclusiveness of urban social spaces, ultimately contributing to the sustainable development of urban communities.”

7.L438:Same as mentioned above. The section mentions various aspects of empirical studies on activity and living spaces. To enhance clarity, you might organize this information into subsections or tables, categorizing the studies by the topics explored (e.g., spatial boundaries, quality evaluation, characteristics, influencing factors).

Thank you very much for your suggestion. I have organized the content of the section into a table for presentation based on your suggestion, thus improving the clarity of the section for readers, and the added table content is as follows:

Table 5 Summary of Empirical Studies on residents' activity space and living space

Research Aspects

Description

Research Topics

- Spatial boundaries identification, quality evaluation, characteristics, influencing factors.

- Daily activity characteristics, behavior-based spatial characteristics, influencing factors analysis.

Data Sources

- Traditional data (in-depth interviews, activity diaries), new data (GPS, cell phone signaling).

Quality Evaluation

- Multiple dimensions of quality (e.g., resource accessibility, satisfaction, sense of belonging).

- Evaluation based on different data sources and dimensions.

Characteristics and Factors

- Analysis of residents' activity spaces and living spaces, identification of influencing factors.

- Socioeconomic conditions, built environment, ICTs, impact of digitalization.

Insights into Non-Mainstream Groups

- Study of low-income clusters, children, older adults, refugees, and their daily activities.

- Mobility analysis, social relationships, social equity, and social inclusion issues.

Urban Planning Implications

- Basis for urban planning, improving residents' well-being, promoting city sustainability.

- Recognition of diversity and challenges in activity and living spaces design.

8.Discuss how the findings from empirical studies on activity and living spaces can inform urban planning and policy decisions aimed at improving residents' well-being and promoting sustainable development. What practical recommendations can be drawn from this research?

       Really thank you very much for your suggestion. Empirical studies on activities and living spaces do provide a wealth of information that is invaluable for urban planning and policy development. I have added the policy implications of the empirical research findings in the section as follows:

       “These findings emphasize the importance of optimizing resource allocation to ensure equitable access to essential services, enhancing housing and community design to create safe and comfortable environments, promoting digital inclusion for all residents, addressing spatial inequalities through flexible zoning and mixeduse development, actively engaging the community in the planning process, adopting an adaptive approach to accommodate evolving trends, and, above all, adopting a holistic and inclusive approach to urban planning. In conclusion, empirical studies on activity and living spaces offer a wealth of information that can be invaluable for urban planning and policymaking. By incorporating these perspectives, policymakers can consider various viewpoints and communities while developing urban policies that enhance the quality of life for residents and facilitate sustainable urban growth.”

9.L446-449:When discussing empirical studies, provide insights into the types of data sources used (e.g., GPS data, cell phone signaling data, interviews, surveys). This information can help readers understand the methodologies employed.

       Thank you very much for your suggestion, this is indeed an area that needs to be added to my article. In response, I have illustrated the main types of data adopted in the empirical research in the subsequent sorting process so that readers can fully understand the research methodology adopted in the relative empirical research, and an example of the additions made is as follows:

       “using a questionnaire survey method to study workers' satisfaction with...”

“a Nanjing-specific study based on activity diaries survey data found that”

 “A study of older adults based on household travel survey data has found that there are two main”

“Through in-depth interviews, they have found that children's outdoor activity space is mainly”

       In addition, in order to give the reader a more detailed understanding of these methods, I have explained them with the following additions:

“In-depth interview is a detailed and targeted conversation conducted by the researcher or research team to understand the respondent's viewpoints, experiences, attitudes and feelings, etc. The questionnaire data are collected by asking a series of questions to the respondents and recording their responses. GPS data and mobile phone signaling data contain a large amount of geospatial information, both of which are different in accuracy, but can dynamically record individual movement characteristics and vividly display the living and activity space of residents [129].”

10.L541:Incorporate specific case studies or examples of how these cutting-edge urban planning concepts have been applied to real-world urban development. This can make these concepts more concrete and relevant to the reader. At the same time, the validity and challenges associated with these cutting-edge concepts can be critically assessed. Should the limitations or criticisms of these approaches be discussed?

Thank you very much for your suggestions. I have provided a brief statement of specific examples of these cutting-edge urban planning concepts so that the reader can get a clearer picture of how these cutting-edge urban planning concepts can be applied to real-world urban development.

       The specific additions are listed below:

“For example, Oswiecim realizes the sustainable development of towns by enriching their functions, designing town spaces based on the concept of balance, and the main measures include the provision of jobs and housing to meet the basic needs of Oswiecim's residents, increasing the accessibility of local sports and recreational venues, good safety monitoring, taking care of the needs of each social group, and encouraging social participation [165]. Japan focuses on public participatory resilience building with the community as the basic unit, and about 30% or more of Kyoto's resilience strategy emphasizes the importance of community and public participation in reducing the risk of future shocks. The Kyoto government encourages citizens to participate in improving the safety and security of their neighborhoods and cities, and builds communication channels between the government and citizens in the community through smart programs to maintain and strengthen the assets of urban facilities while also forming a good government-society partnership [153]. The construction of smart cities has also received attention from many countries, and smart governance, smart people's livelihood, and solving problems such as traffic congestion, environmental pollution, and resource constraints are the development goals of smart cities. Shanghai is one of the earliest pilot cities in China to build a smart city, and Shanghai has made great progress in information infrastructure construction, information perception and intelligent application, new generation of information technology industry and information security assurance. etc. [166]. The sixth phase of the European Healthy Cities Program supports cities to strengthen linkages, bring key stakeholders together for health and wellbeing, and use leadership, innovation and change to increase the potential to address local public health challenges [159]. It can be seen that many countries are applying these concepts to urban development planning to create more sustainable, safe and healthy urban environments.”

Meanwhile, I strongly agree with your point of exploring the limitations of these concepts. Therefore, I have thought about discussing the limitations of each of the major cutting-edge urban planning concepts, with the following additions:

“However, there are limitations to the application of these urban planning concepts, such as the need for largescale investment in the construction of resilient cities; the construction of smart cities, which may bring about issues of privacy and security, digital divide and technological dependency; the construction of healthy cities, which may also lead to issues of health inequality; and the planning of sustainable cities, which requires weighing conflicting interests, such as those of the environment, society and the economy. Therefore, In applying these cutting-edge concepts, urban planners and governments need to develop integrated policies that encourage innovative, diverse solutions and ensure equitable, inclusive and sustainable urban development.”

Thanks again for such detailed and pertinent advice, it helps me a lot.

Reviewer 3 Report

Comments and Suggestions for Authors

Abstract - suggested to do editing and some addition since it lacks the elements that usually have in the abstract (where are the problem statement, objective of research, methodology and research achievement)

This paper is a literature review and a well written discussion on the scholarly views of the past papers/studies. However, it did not mention it is a fundamental or a review paper.

The title of this paper should be modified as to show it is a review paper.

I would suggest this paper include several tables as to distinguish and clarify all the past studies that have been selected and then elaborate on it.

In page 3, it stated that there are several different theoretical perspectives, however, it failed to show or highlight that theoretical perspectives. that's why, it is better to produce a table and highlight all the relevant and appropriate reviews/past studies/theories.

Author Response

  1. Abstract - suggested to do editing and some addition since it lacks the elements that usually have in the abstract (where are the problem statement, objective of research, methodology and research achievement)

Thank you very much for your advice and I apologize that my previous summary was not exhaustive enough. I have revised the abstract to include a description of the problem, research objectives, and the impact of the paper. Additionally, I have emphasized in the abstract that this article is a literature review.

Revised abstract: “Urban social space and sustainable urban development are both prominent areas of research in urban studies. The development of a city is closely tied to the development of its social space. The level of sustainable development in a city can be assessed by examining the evolution of its urban social space. Therefore, the relationship between the two is highly interconnected. However, the social dimension of sustainable development has always received the least attention compared to the economy and the environment. Therefore, this paper examines urban transformation and urban spatial structure, social composition and integration of residents in urban communities, the activity space and living space of urban residents, and the social integration of urban residents, activity space and living space, and urban planning, this paper provides a comprehensive literature review of research on urban social space and sustainable development. On the one hand, this literature review thoroughly examines the correlation between urban social space and sustainable urban development. On the other hand, it broadens the perspectives of urban research, highlights the key role of social dimensions in sustainable urban development, and helps to draw the attention of academics to this topic. In addition, this literature review may provide policymakers with more comprehensive information on urban social space and sustainable development.”

  1. This paper is a literature review and a well written discussion on the scholarly views of the past papers/studies. However, it did not mention it is a fundamental or a review paper.

Thank you for your advice. This was indeed an oversight on my part. I have addressed the issue by explicitly stating that this is a review paper in various sections of the article, including the abstract and introduction.

For example, "This paper provides a comprehensive literature review..." in the abstract, and "This review paper is meticulously structured as...".

  1. The title of this paper should be modified as to show it is a review paper.

Thank you for your suggestion. The title of this paper, "Research Progress and Prospect," already indicates that it is a literature review. However, you are correct that this approach is not direct enough. Therefore, I have revised the title to "Review of Research on Urban Socio-Spatial and Sustainable Development."

  1. I would suggest this paper include several tables as to distinguish and clarify all the past studies that have been selected and then elaborate on it.

Thank you very much for your suggestion. I have added tables to the main content section of the article to provide readers with a clearer understanding of the theoretical and empirical studies in the field.

For example: “

Research Aspects

Description

Research Topics

Data Sources

Quality Evaluation

- Spatial boundaries identification, quality evaluation, characteristics, influencing factors.

- Daily activity characteristics, behavior-based spatial characteristics, influencing factors analysis.

- Traditional data (in-depth interviews, activity diaries), new data (GPS, cell phone signaling).

- Multiple dimensions of quality (e.g., resource accessibility, satisfaction, sense of belonging).

- Evaluation based on different data sources and dimensions.

Characteristics and Factors

- Analysis of residents' activity spaces and living spaces, identification of influencing factors.

- Socioeconomic conditions, built environment, ICTs, impact of digitalization.

Insights into Non-Mainstream Groups

- Study of low-income clusters, children, older adults, refugees, and their daily activities.

- Mobility analysis, social relationships, social equity, and social inclusion issues.

Urban Planning Implications

- Basis for urban planning, improving residents' well-being, promoting city sustainability.

- Recognition of diversity and challenges in activity and living spaces design.

Research Aspects

Description

  1. In page 3, it stated that there are several different theoretical perspectives, however, it failed to show or highlight that theoretical perspectives. that's why, it is better to produce a table and highlight all the relevant and appropriate reviews/past studies/theories.

       Thank you very much for your suggestion. I have added the table at the appropriate place in the article as follows:

Theoretical  Perspectives

Description

Chicago School & Urban Ecology

- The Chicago School's urban ecology theory views the city as an organism with socio-spatial structures shaped by ecological processes.

- Burgess's concentric circle model divides the city into functional areas through concentric circles.

- Later models, like the sector and multiple nuclei models, expanded upon and refined the original theory.

Dialectical Analysis

- Analyzes urban socio-spatial transformation from a metaphysical and historical perspective.

Feminist-Materialist Theory

- Focuses on anti-racist feminist-materialist urban theory, emphasizing care, the common, and the collective in urban socio-spatial transformation.

Neoliberalism

- Analyzes urban socio-spatial transformation within the context of neoliberalism.

Social Engineering

- Examines government-dominated cities where social engineering ideas and a "engineering type of mind" play a significant role in transformation.

Comments on the relevant theories have already been mentioned in the summary section of the article, as follows: “It is not difficult to find that the existing studies focusing on the transformation and reconstruction of urban social space have diversified themes, both in terms of theoretical discussion and analysis of practical problems. However, a significant shortcoming is that the existing studies are mostly centered on corroborating or questioning the existing theories, and no new theories have been created.”

Reviewer 4 Report

Comments and Suggestions for Authors

Thank you very much for sending the work. It is an extensive and robust work, to which I would like to make some suggestions.

It is necessary to systematise in a table and a sequential scheme the review of the state of the art with emphasis on the categories

Author Response

1.It is necessary to systematise in a table and a sequential scheme the review of the state of the art with emphasis on the categories

       Thank you very much for your suggestion. I have added 5 tables to the article to provide readers with a clearer understanding of the existing theories and empirical studies in the field.

Round 2

Reviewer 1 Report

Comments and Suggestions for Authors

Dear authors, 

Almost all my comments have been addressed. While I agree with you that providing an in-depth discussion of the other two pillars of economic and ecological sustainability in the study is not appropriate, I still strongly recommend the authors provide a paragraph discussing these aspects to make sure sustainability is adequately covered. You might suggest further studies to focus on these two aspects. Please refer to my previous comment per suggestion.

Comments on the Quality of English Language

Good.

Author Response

(x) Minor editing of English language required

Thank you very much for your advice, I have checked and revised the article again for presentation, grammar, etc., and all changes have been marked.

Comments and Suggestions for Authors

Dear authors, 

Almost all my comments have been addressed. While I agree with you that providing an in-depth discussion of the other two pillars of economic and ecological sustainability in the study is not appropriate, I still strongly recommend the authors provide a paragraph discussing these aspects to make sure sustainability is adequately covered. You might suggest further studies to focus on these two aspects. Please refer to my previous comment per suggestion.

Thank you very much for your suggestion, I agree with you very much on this one. I really should have made a brief statement on economic and ecological sustainability to make the concept of sustainability complete.

In response, I have added a description of the concepts and main research elements of environmental sustainability and economic sustainability in the introduction section. The additions are listed below:

“Consequently, the academic community has shown significant interest in the sus-tainable development of the environment. [9] Rapid urban development has caused extensive ecological damage, harming the resources on which human beings depend for survival, such as water, atmosphere, and soil [8]. Therefore, the protection and management of the ecological environment are the focus and hotspot in the study of environmental sustainability. Programs such as developing a low-carbon and green economy [10], promoting recycling and urban waste classification [11], and urban planning based on an ecological perspective [12] have been proposed. Research on measuring the effectiveness of environmental protection and governance policies, such as evaluating ecological sustainability policies [9] and assessing the sustainability of urban ecosystems [13], is also a crucial area of study in environmental sustainability research.

Economic sustainability focuses on the maintenance of capital resources, includ-ing manufacturing capital, natural capital, and financial capital [14]. It emphasizes that economic growth should not jeopardize the economic prospects of future genera-tions [15]. Existing studies focus on the factors that impact economic sustainability, such as the level of innovation [16] and the digital economy [17]. In addition, studies on economic sustainability are often intertwined with ecology, with a focus on topics such as the ecological impact on sustainable economic development [18], and the en-vironmental assessment of economic development [19].”

Additionally, I emphasized in section 6.2 of the paper that future research should focus on the interplay of the three pillars of sustainability. The specifics of the additions are listed below:

“In addition, sustainable urban development requires proper coordination among the social, environmental, and economic dimensions. Therefore, future research needs to pay more attention to the interactions among these three pillars in order to find more effective ways to promote coordination and synergies among them, ultimately achieving sustainable development in cities at all levels.”

Reviewer 2 Report

Comments and Suggestions for Authors

The paper has been effectively revised according to the opinions of reviewers, and I think the revised paper meets the requirements for publication.

However, The conclusion part should give more specific directions for the guidance of future research.

Fig1 and Fig 2 have very low resolution and needs to be replaced.

Comments on the Quality of English Language

Minor editing of English language required

Author Response

(x) Minor editing of English language required

Thank you very much for your advice, I have checked and revised the article again for presentation, grammar, etc., and all changes have been marked.

Comments and Suggestions for Authors

The paper has been effectively revised according to the opinions of reviewers, and I think the revised paper meets the requirements for publication.

However, The conclusion part should give more specific directions for the guidance of future research.

Thank you very much for your advice; it is very useful to me. My previous conclusion section was not sufficiently specific about future research directions. In response, I have added a section to the conclusions that highlights future research directions. The addition is as follows:

“6.2 future research directions

Despite the extensive research conducted in the field of urban social space and sustainable development, there are still some shortcomings. Moving forward, future research and practice should aim to better achieve the goal of sustainable urban development.

At the city level, new theories of urban socio-spatial reconfiguration need to be actively explored in order to adapt to the ever-changing urban social structure. These new theories should be able to better address the transition from homogeneous to heterogeneous urban structures, and offer more practical solutions to the social problems caused by urban transformation.

At the community level, future research should continue to focus on interactions and cooperation among residents, including those from different age groups, ethnicities, and socioeconomic backgrounds. Additionally, research should explore strategies for building more inclusive communities. Focusing on community governance and resident participation, it is important to explore and utilize the role of digital technology in community interaction as much as possible. These future research directions will contribute to a more comprehensive understanding of sustainable development issues at the community level and provide more focused recommendations for community planning and policy making.

At the resident level, it is important to prioritize the needs of vulnerable groups, such as the elderly and the poor, in order to ensure that all social groups have access to the resources and opportunities offered by the city. Programs will be sought to enhance the well-being of all residents, with a focus on their living space and mobility. In addition, the study can explore the relationship between sustainable lifestyles and residents' behaviors. How to encourage and support residents to adopt more environmentally friendly lifestyles, including energy conservation, waste reduction, and sustainable transportation, is an area of research closely related to sustainable development.

At the planning level, it is important to continue exploring ways in which urban planning can make a real difference. This may require broader public participation to ensure that the planning process reflects the needs and perspectives of diverse social groups.

In addition, sustainable urban development requires proper coordination among the social, environmental, and economic dimensions. Therefore, future research needs to pay more attention to the interactions among these three pillars in order to find more effective ways to promote coordination and synergies among them, ultimately achieving sustainable development in cities at all levels.

In conclusion, through ongoing in-depth research conducted at the city, community, and resident levels, and by applying this research to urban planning practices, we can expect to achieve a more optimized urban spatial structure, more inclusive communities, and happier residents' lives. This, in turn, will promote sustainable urban development. This will require continuous innovation and interdisciplinary collaboration to address the complex challenges of urban socio-spatial and sustainable development.”

Fig1 and Fig 2 have very low resolution and needs to be replaced.

        Thank you very much for your suggestion. The resolution of Figure 1 and 2 is indeed too low. I have made adjustments, and the resolution of the two images after the adjustment is 200.

Reviewer 3 Report

Comments and Suggestions for Authors

Appropriately written as suggested to the comments before this.

Author Response

(x) English language fine. No issues detected

Comments and Suggestions for Authors

Appropriately written as suggested to the comments before this.

Thank you very much for your advice, I have checked and revised the article again for presentation, grammar, etc., and all changes have been marked. In addition, I have made changes to each of the suggestions you made in the previous round. These include adding improvements to the abstract, emphasizing that this is a review paper, revising the title, and adding tables to provide a clearer overview of the research that has been conducted. Thank you again for your suggestions. If there are still changes that need to be made, I would appreciate it if you could point them out.